DATA RELEASE

# A chromosome-level genome assembly and annotation of the maize elite breeding line Dan340

Yikun Zhao[1,†], Yuancong Wang[2,†], De Ma[3,†], Guang Feng[4], Yongxue Huo[1], Zhihao Liu[1], Ling Zhou[2], Yunlong Zhang[1], Liwen Xu[1], Liang Wang[4], Han Zhao[2], Jiuran Zhao[1,*] and Fengge Wang[1,*]

1 Maize Research Center, Beijing Academy of Agricultural and Forest Sciences (BAAFS)/Beijing Key Laboratory of Maize DNA Fingerprinting and Molecular Breeding, Beijing 100097, , China
2 Provincial Key Laboratory of Agrobiology, Institute of Crop Germplasm and Biotechnology, Jiangsu Academy of Agricultural Sciences, Nanjing 210014, , China
3 Novogene Bioinformatics Institute, Beijing 100015, China
4 Dandong Academy of Agricultural Sciences, Liaoning 118199, China

## ABSTRACT

**Background:** Maize is an important model organism for genetics and genomics research. Though reference genomes of maize are available, some genomes of important genetic germplasms for maize breeding are still lacking, for instance, the cultivar Dan340, which is a backbone inbred line of the LvDa Red Cob Group with several desirable characteristics. In this study, we constructed a high-quality chromosome-level reference genome for Dan340 by using long HiFi reads, short reads, and Hi-C. The final assembly of the Dan340 genome was 2348.72 Mb, which was anchored to 10 chromosomes. Repeat sequences accounted for 73.40% of the genome and 39,733 protein-coding genes were annotated. Comparative genomic analysis between Dan340 and other maize lines identified that 1806 genes from 359 gene families were specific to Dan340.
**Conclusions:** Our genome assembly and annotation provide a valuable resource for improving maize breeding and further understanding the intraspecific genome diversity in maize.

**Subjects** Genetics and Genomics, Agriculture, Plant Genetics

**Submitted:** 12 January 2022

* Corresponding authors. E-mail: maizezhao@126.com; fenggewangmaize@126.com

† Contributed equally.

Preprint submitted at https://doi.org/10.1101/2021.04.26.441299

## DATA DESCRIPTION

### Background

Maize (*Zea mays* ssp. *mays* L., NCBI:txid381124) is one of the most important crops grown worldwide for food, forage, and biofuel, with an annual production of more than 1 billion tons [1]. Owing to the rapid human population growth and economic demand, maize has been predicted to account for 45% of the total cereal demand by 2050 [2]. In addition, it is an important model organism for fundamental research in genetics and genomics [3].

Because of its importance in crop science, genetics and genomics, several reference genomes of common maize inbred lines used in breeding have been released since 2009 [4–8]. However, comparative genomic analyses have found that maize genomes exhibit high levels of genetic diversity among different inbred lines [1, 7, 9]. Meanwhile, accumulating studies have suggested that one or a few reference genomes cannot fully represent the genetic diversity of a species [7, 10, 11].

The maize cultivar Dan340 is an excellent backbone inbred line of the LvDa Red Cob Group that has several desirable characteristics, such as disease resistance, lodging

resistance, high combining ability, and wide adaptability. More than 50 maize hybrid breeds have been derived from Dan340 since 2000, and their planting area has reached 19 million ha. It is considered that Dan340 originated from a landrace in China and exhibits significant genetic differences from other maize germplasms that represent the most important core maize germplasms in China [12]. Therefore, Dan340 could serve as a model inbred line for the genetic dissection of desirable agronomic traits, combining ability, heterosis, and breeding history.

In the present study, we constructed a high-quality chromosome-level reference genome for Dan340 by combining PacBio long HiFi sequencing reads, Illumina short reads, and chromosomal conformational capture (Hi-C) sequencing reads. The completeness and continuity of the resulting genome are comparable with those of other important maize inbred lines: B73 [4], Mo17 [7], SK [13], PH207 [5], and HZS [8]. Furthermore, comparative genomic analyses were performed between Dan340 and other maize lines. Genes and gene families specific to Dan340 were identified. In addition, large numbers of structural variations between Dan340 and other maize inbred lines were detected. The assembly and annotation of this genome will increase our understanding of the intraspecific genomic diversity in maize and provide a novel resource for maize breeding improvements.

## Plant materials and DNA sequencing

The inbred line Dan340 (Figure 1) was selected for genome sequencing and assembly because it is an elite maize cultivar that plays an important role in maize breeding and genetic research. The plants were grown at 25 °C in a greenhouse of the Beijing Academy of Agriculture and Forestry Sciences, Beijing, China. Fresh and tender leaves were harvested from the best-growing individual, immediately frozen in liquid nitrogen, and then preserved at −80 °C in the laboratory prior to DNA extraction. Genomic DNA was extracted from the leaf tissue of a single plant using the DNAsecure Plant Kit (Tiangen Biotech Co., Ltd., Beijing, China). To ensure that the DNA extracts were useable for all types of genomic libraries, their quality and quantity were evaluated using a NanoDrop 2000 spectrophotometer (NanoDrop Technologies, Wilmington, DE, USA) and electrophoresis on a 0.8% agarose gel, respectively.

In recent years, third-generation DNA sequencing technologies have undergone rapid technological innovation and are now widely used in genome assembly. In this study, PacBio circular consensus sequencing (CCS) libraries were prepared using the SMRTbell Express Template Prep Kit 2.0 (Pacific Biosciences, Menlo Park, CA, USA; Ref. No. 101-685-400), following the manufacturer's protocols, and they were subsequently sequenced on the PacBio sequel II platform (Pacific Biosciences, RRID:SCR_017990). As a result, 63.53 Gb (approximately 27× coverage) of HiFi reads was generated and used for the genome assembly.

In addition, one Illumina paired-end sequencing library, with an insert size of 350 bp, was generated using the NEB Next Ultra DNA Library Prep Kit (NEB, Ipswich, MA, USA) following the manufacturer's protocol and then sequenced using an Illumina HiSeq X Ten platform (Illumina, San Diego, CA, USA, RRID:SCR_016385) at the Novogene Bioinformatics Institute, Beijing, China. Approximately 80.66 Gb (~34×) of Illumina sequencing data were obtained.

One Hi-C library was constructed using young leaves following previously published procedures [14], with slight modifications outlined in our published protocol [15] (Figure 2).



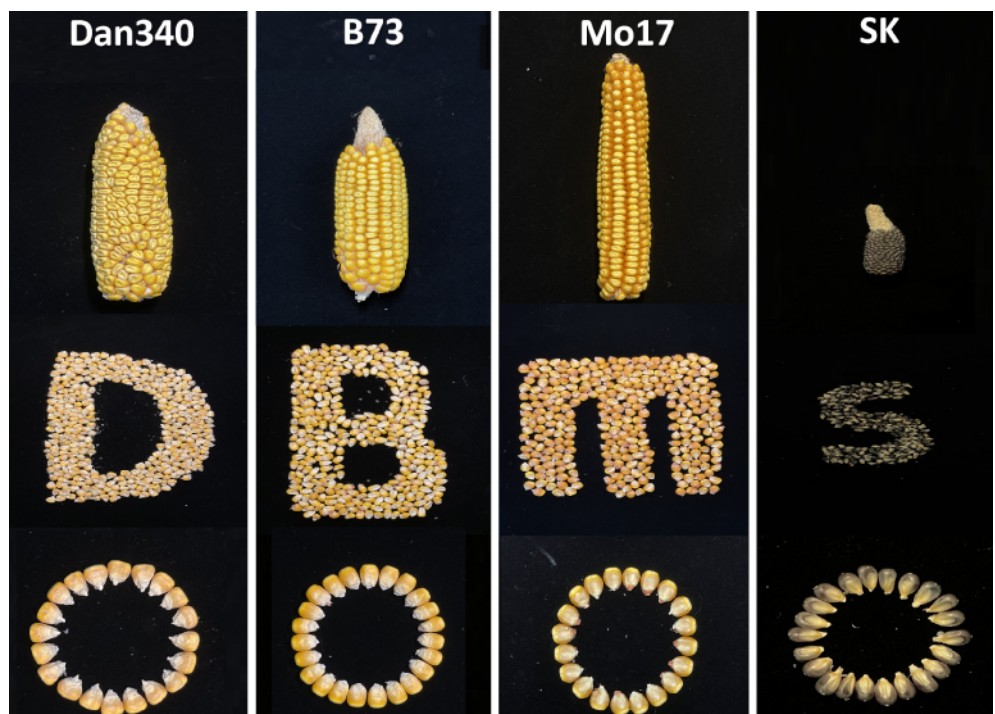

**Figure 1.** Ear appearances of the maize inbred lines Dan340, B73, Mo17, and SK.

In brief, approximately 5-g leaf samples from seedlings were cut into minute pieces and cross-linked using a 4% formaldehyde solution at room temperature in a vacuum for 30 min. Each sample was mixed with excess 2.5 M glycine for 5 min to quench the cross-linking reaction and then placed on ice for 15 min. The cross-linked DNA was extracted and then digested for 12 h with 20 units of *DpnII* restriction enzyme (NEB, Ipswich, MA, USA, Catalog #R0543S) at 37 °C. Next, the resuspended mixture was incubated at 62 °C for 20 min to inactivate the restriction enzyme. The sticky ends of the digested fragments were biotinylated and proximity ligated to form enriched ligation junctions and then ultrasonically sheared to a size of 200–600 bp. The biotin-labelled DNA fragments were pulled down and ligated with Illumina paired-end adapters, and then amplified by PCR to produce the Hi-C sequencing library. The library was sequenced using an Illumina HiSeq X Ten platform with 2 × 150 bp paired-end reads. After removing low-quality sequences and trimming adapter sequences, 304.37 Gb (approximately 130×) of clean data were generated and used for the genome assembly.

## Genome assembly

To obtain a high-quality genome assembly of Dan340, we employed both PacBio HiFi reads and Illumina short reads, with scaffolding informed by high-throughput Hi-C. The assembly was performed in a stepwise fashion. First, a *de novo* assembly of the long CCS reads generated from PacBio single-molecule real-time (SMRT) sequencing was performed using Hifiasm [16] (RRID:SCR_021069). A total of two SMRT cells produced 4,073,418 subreads, with an average length of 15,598 bp and a read N50 of 15,715 bp. The generation of HiFi reads and adapter trimming was performed using PacBio SMRTLink (Version 8.0) [17] with

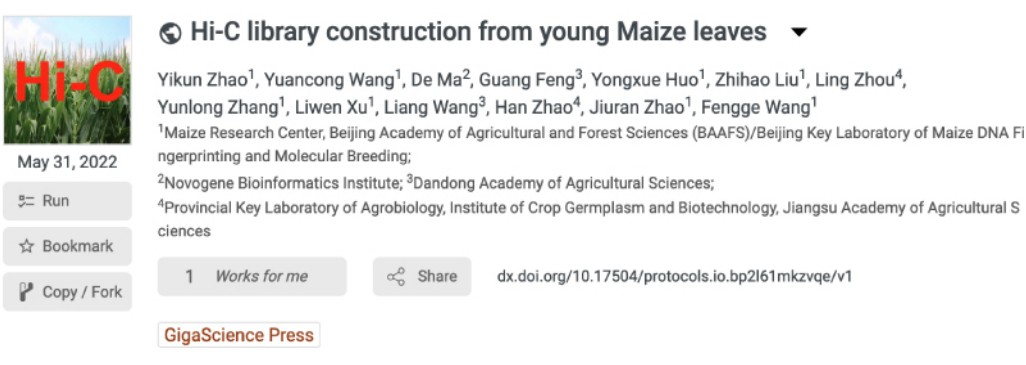

**Figure 2.** Protocols.io protocol for the Hi-C library construction from young Maize leaves [15]. https://www.protocols.io/widgets/doi?uri=dx.doi.org/10.17504/protocols.io.bp2l61mkzvqe/v1

default parameters, followed by the deduplication of reads using pbmarkdup (Version 0.2.0) [18], as recommended by PacBio. Next, HiFi reads were aligned to each other and assembled into genomic contigs using Hifiasm [16] with default parameters. Next, the primary contigs (p-contigs) were polished using Quiver [19] by aligning the SMRT reads. Then, Pilon [20] (RRID:SCR_014731) was used to perform the second round of error correction using the short paired-end reads generated by the Illumina Hiseq platforms. Subsequently, the Purge Haplotigs pipeline [21] was used to remove redundant sequences formed due to heterozygosity. The draft genome assembly was 2348.68 Mb; it reached a high level of continuity and a contig N50 length of 45.11 Mb.

To reduce Hi-C reads having a bias due to experimental artefacts, we removed the following read types using HiCUP [22] (RRID:SCR_005569): (a) reads with ≥10% unidentified nucleotides (N); (b) reads with >10 nt aligned to the adapter, allowing ≤10% mismatches; and (c) reads with >50% bases having a Phred quality <5. Next, the filtered Hi-C reads were aligned against the contig assemblies using BWA (Version 0.7.8, RRID:SCR_010910) [23]. Reads were excluded from subsequent analyses if they did not align within 500 bp of a restriction site or did not uniquely map. Also, the number of Hi-C read pairs linking each scaffold pair was tabulated. ALLHiC (Version 0.8.12) [24] was used in simple diploid mode to scaffold the genome and optimize the ordering and orientation of each clustered group, producing a chromosome-level assembly. The Juicebox Assembly Tools (Version 1.9.8, RRID:SCR_021172) [25] were used to visualize and manually correct the large-scale inversions and translocations to obtain the final pseudo-chromosomes (Figure 3). Finally, 2315 scaffolds (representing 91.30% of the total length) were anchored to 10 chromosomes (Figure 4).

The final assembly of the Dan340 genome was 2348.72 Mb, including 2738 contigs and 2315 scaffolds, with N50 of 41.49 Mb and 215.35 Mb, respectively (Table 1).

## Evaluation of the assembly quality

We assessed the quality of the assembly using several independent methods. First, the short reads obtained from the Illumina sequencing data were aligned to the final assembly using BWA [26]. Our results showed that the percent of reads mapped to the reference genome was 97.48%. Second, a total of 248 conservative genes existing in six eukaryotic model organisms were selected to form the core gene library for the Core Eukaryotic Genes

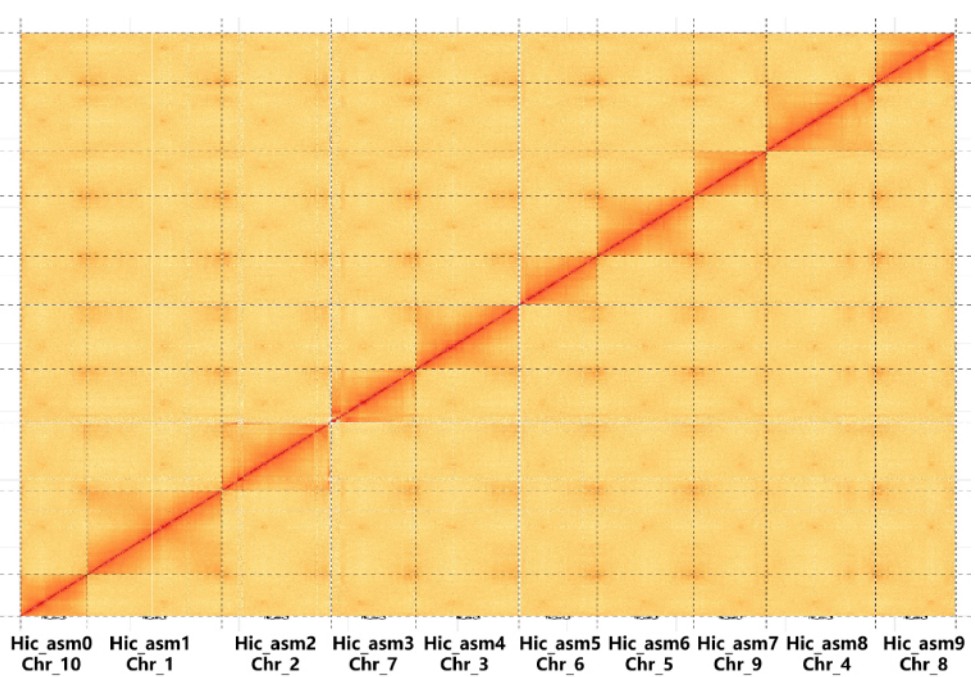

Figure 3. Hi-C contact heat map displaying the inter- and intra-chromosomal interactions in the genome of the maize inbred line Dan340.

Table 1. Genome assembly and annotation statistics for the four tested maize inbred lines.

| Genomic features | Dan340 | B73 | Mo17 | SK |
|---|---|---|---|---|
| Assembled genome size (bp) | 2,348,678,871 | 2,182,075,994 | 2,104,465,715 | 2,161,392,594 |
| Number of scaffolds | 2315 | 687 | 2203 | 671 |
| Total length of scaffolds (Mb) | 2348.72 | 2182 | 2182 | 2162 |
| Scaffold N50 | 222,765,871 | 226,353,449 | 220,382,597 | 73,237,962 |
| Number of contigs | 2738 | 1395 | 9040 | 1090 |
| Total length of contigs (Mb) | 2,144,444 | 2,178,268 | 2,147,495 | 2,150,874 |
| Contig N50 | 45,109,016 | 47,037,903 | 1,491,782 | 15,776,512 |
| Number of genes | 39,733 | 39,756 | 38,620 | 43,271 |

Mapping Approach (CEGMA) [27] (RRID:SCR_015055) evaluation. To evaluate its integrity, our assembled Dan340 genome was aligned to this core gene library using TBLASTN (RRID:SCR_011822) [28], GeneWise (Version 2.2.0, RRID:SCR_015054) [29], and the GeneID tools (Version 1.4 RRID:SCR_021639) [30]. Our results showed that 238 complete (95.97%) and 243 partial (97.98%) genes were detected in our assembly. Third, the completeness was assessed using the benchmarking universal single-copy orthologs (BUSCO) [31] (RRID:SCR_015008). The final assembly was tested against BUSCO (v.3) with embryophyta_odb10 database [32], which includes 1614 conserved core genes. Our results showed that 98.08% (1583), 1.11% (18), and 0.81% (13) of the plant single-copy orthologs were present in the assembled Dan340 genome as complete, fragmented, or missing genes, respectively. Fourth, the long-terminal repeat (LTR) Assembly Index (LAI) metric was used to evaluate the assembly continuity in Dan340 and three other maize genomes (B73, Mo17, and SK; Figure 5). The intact LTR retrotransposons were identified in the four genomes using LTRharvest (Version 1.6.1, RRID:SCR_018970) [33], LTR_Finder (Version 1.07,

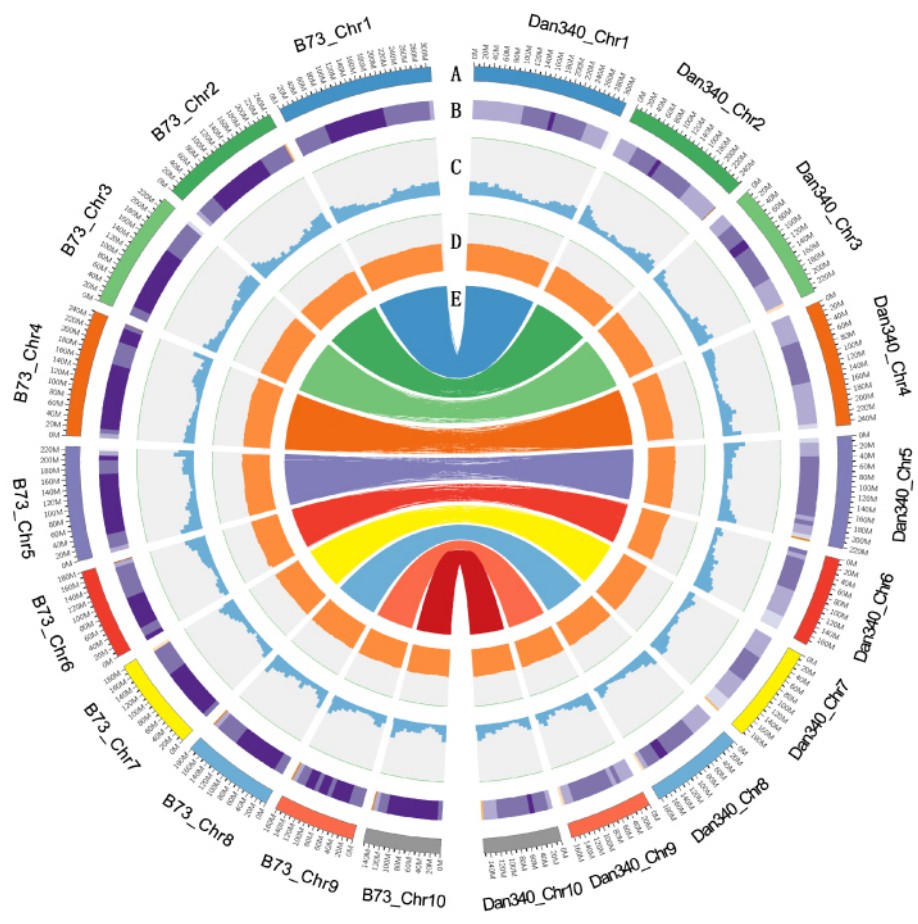

**Figure 4.** Circos plot of genomic features. Outer-to-inner tracks indicate the following: (A) Chromosome numbers of Dan340 and B73; (B) Repeat density; (C) Histogram of gene density distributions along the chromosomes; (D) Histogram of GC content distributions along the chromosomes; (E) Syntenic relationships of gene pairs between Dan340 and B73 genomes identified using the best-hit method.

**Table 2.** LAI scores of the four tested maize inbred lines.

| Lines | Dan340 | B73 | Mo17 | SK |
|---|---|---|---|---|
| LAI | 25.13 | 24.94 | 24.45 | 27.12 |

RRID:SCR_015247) [34], and LTR_retriever (Version 2.9.0, RRID:SCR_017623) [35]. The LAI pipeline was executed using the following parameter settings: -t 20 -intact genome.fasta.pass.list -all genome.ltr.fasta.out. Our Dan340 genome had an LAI score of 25.13, which was relatively high among the four maize genomes compared in this study. B73, Mo17, and SK scored 24.94, 24.45, and 27.12, respectively (Figure 5 and Table 2). A higher LAI score indicates a more complete genome assembly because more intact LTR retrotransposons are identified, as was the case of our Dan340 genome. Furthermore, whole-genome sequence alignments of Dan340 to the genomes of the other three maize inbred lines demonstrated that our assembly has highly collinear relationships with other published maize genomes (Figure 6). Taken together, our assessment results suggest that the Dan340 genome assembly is of high quality.

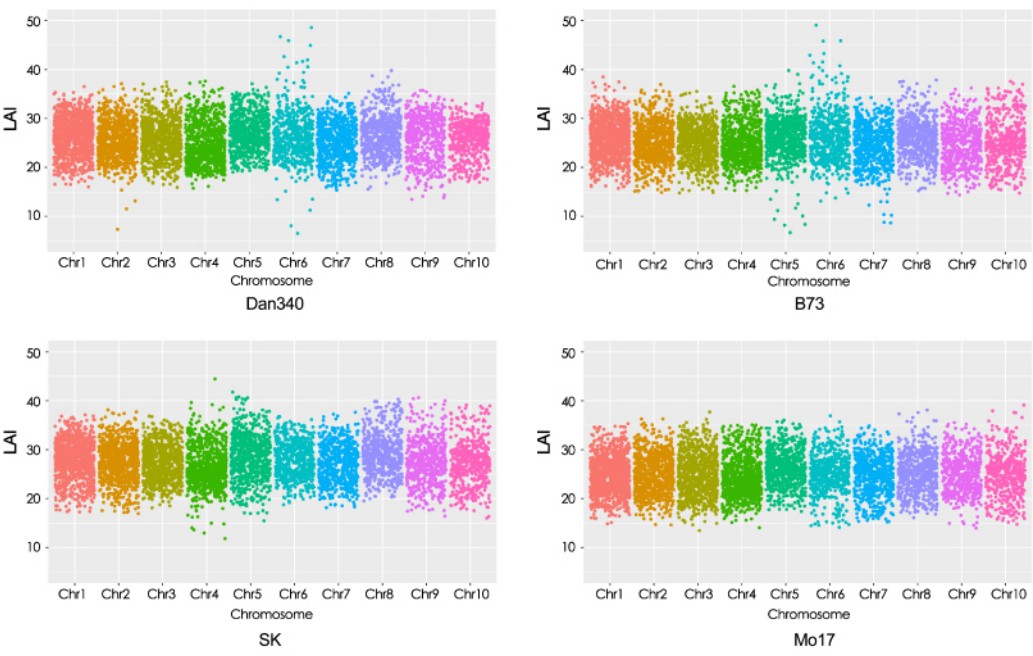

**Figure 5.** Genome-wide LAI scores for Dan340, B73, Mo17, and SK.

## Genome annotation

Repeat sequences of the Dan340 genome were annotated using both *ab initio* and homolog-based search methods. For the *ab initio* prediction, RepeatModeler (Version 1.0.8, RRID:SCR_015027) [36], RepeatScout (Version 1.0.5, RRID:SCR_014653) [37], and LTR_Finder [34] were used to discover transposable elements (TEs) and to build a TEs library. An integrated TEs library and a known repeat library (Repbase Version 15.02, homolog-based, RRID:SCR_021169) were subjected to RepeatMasker (Version 3.3.0 RRID:SCR_012954) [38] to predict the TEs. For the homolog-based predictions, RepeatProteinMask was performed to detect the TEs in our genome by comparing it against a TE protein database. Tandem repeats were ascertained in the genome using Tandem Repeats Finder (Version 4.07b, RRID:SCR_022193) [39]. As a result, 1723.99 Mb of repeat sequences were identified, accounting for 73.40% of the genome size. Among these repeat sequences, 1555.57 Mb were predicted to be long-terminal repeat (LTR) retrotransposons, and 44.53 Mb were predicted to be DNA transposons, accounting for 66.23% and 1.60% of the genome, respectively. Furthermore, among the LTR retrotransposons, the Gypsy and Copia superfamilies comprised 23.81% and 12.75% of the genome, respectively. Thus, retrotransposons accounted for a large proportion of the Dan340 genome, which was consistent with the genomic characteristics of other maize inbred lines (Table 2).

All repetitive regions except the tandem repeats were soft-masked for protein-coding gene annotations. Five *ab initio* gene prediction programs, Augustus (Version 3.0.2, RRID:SCR_008417) [40–42], GENSCAN (Version 1.0, RRID:SCR_013362) [43], GeneID [30], GlimmerHMM (Version 3.0.2, RRID:SCR_002654) [44], and SNAP (Version 2013-02-16, RRID:SCR_007936) [45], were used to predict genes. In addition, the protein sequences of five homologous species (*Sorghum bicolor*, *Setaria italica*, *Hordeum vulgare*, *Triticum aestivum*, and *Oryza sativa*) were downloaded from Ensembl and NCBI. Homologous sequences were

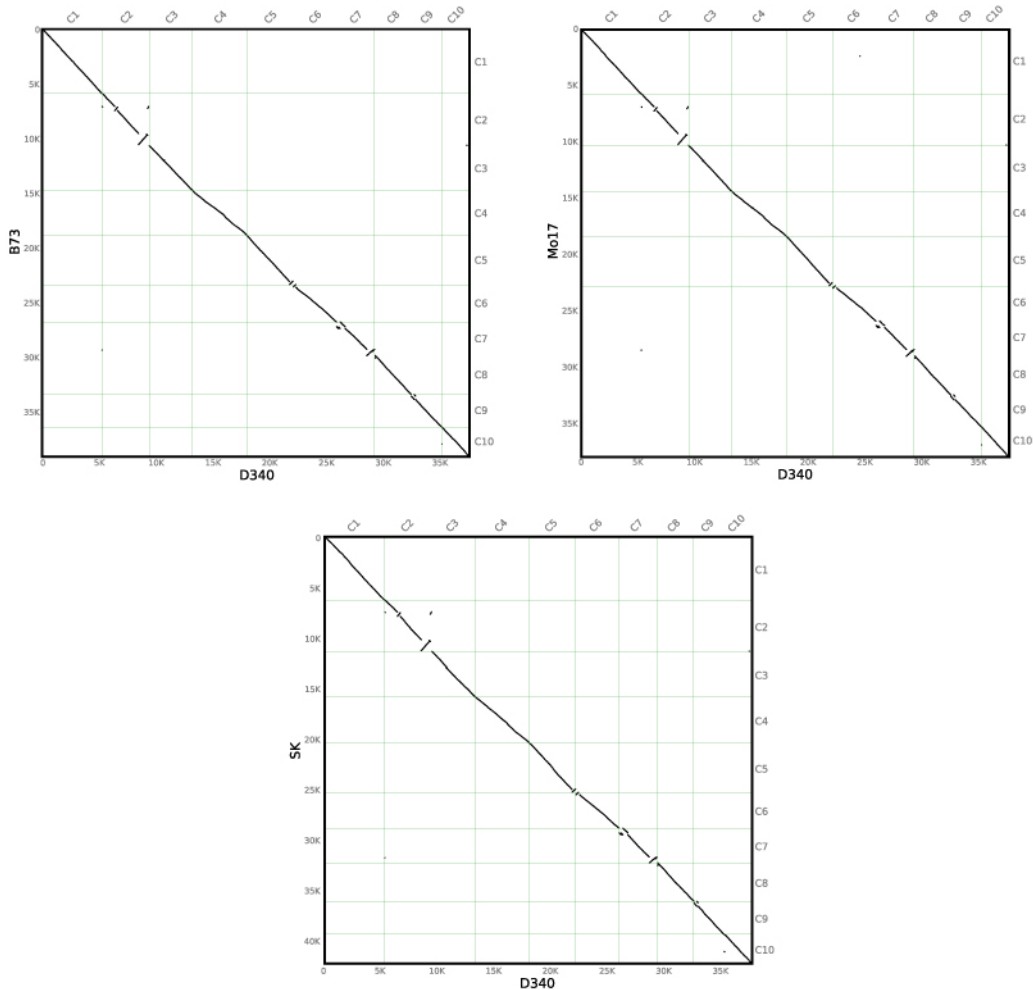

**Figure 6.** (A)–(C) Pairwise comparison of genome sequences using a dot plot between the Dan340 line and B73 (23,350 gene pairs), Mo17 (21,913 gene pairs), or SK (23,016 gene pairs). The horizontal axis represents the target species; the vertical axis represents the reference species; C1–C10 represents the respective chromosomes 1–10; 0–35 k represents the chromosome length scale marks, which mainly reflect the lengths of the chromosomes; a point represents a pair of shared genes.

aligned against the genome using TBLASTN ($E$-value $1 \times 10^{-5}$). GeneWise [29] was employed to predict gene models based on the sequence alignment results.

For the RNA-seq predictions, fresh samples of six tissues (stem, endosperm, embryo, bract, silk, and ear tip) were collected. The total RNA was extracted from each sample using an RNAprep Pure Plant Kit (Tiangen Biotech Co., Ltd., Beijing, China). The isolated, purified RNA, having fragment lengths of approximately 300 bp, was the template for constructing a cDNA library. The NEBNext Ultra RNA Library Prep Kit from Illumina (New England Biolabs, Ipswich, MA, USA) was used to construct the cDNA library following the manufacturer's instructions. The sequencing was performed on an Illumina HiSeq X Ten platform, and 150-bp paired-end reads were generated. Raw reads were trimmed by removing the adapter sequences, reads with more than 5% of unknown base calls (N), and low-quality bases (base quality less than 5). Clean paired-end reads were aligned to the genome using TopHat (Version 2.0.13, RRID:SCR_013035) [46] to identify exon regions and

**Table 3.** Summary statistics of annotated protein-coding genes in Dan340 and other maize inbred lines and common crop species.

| Species | Number | Average transcript length (bp) | Average CDS length (bp) | Average exons per gene | Average exon length (bp) | Average intron length (bp) |
|---|---|---|---|---|---|---|
| Dan340 | 39,733 | 3793.47 | 1140.91 | 4.69 | 243.47 | 719.61 |
| B73 | 39,756 | 3511.78 | 1102.11 | 4.58 | 240.64 | 673.10 |
| Mo17 | 38,620 | 3362.68 | 1140.26 | 4.69 | 242.98 | 601.83 |
| SK | 42,942 | 3857.18 | 1179.17 | 4.83 | 243.93 | 698.48 |
| Hvu | 24,286 | 2116.13 | 1093.77 | 4.1 | 267.02 | 330.19 |
| Osa | 35,679 | 2165.58 | 991.55 | 3.78 | 262.57 | 422.87 |
| Sbi | 34,008 | 2626.44 | 1164.14 | 4.31 | 270.09 | 441.76 |
| Sit | 27,233 | 2982.22 | 1336.29 | 5.14 | 260.2 | 397.98 |
| Tae | 103,539 | 3087.61 | 1277.31 | 4.51 | 283.23 | 515.78 |

Abbreviations: Hvu: *Hordeum vulgare*; Osa: *Oryza sativa*; Sbi: *Sorghum bicolor*; Sit: *Setaria italica*; Tae: *Triticum aestivum*.

splice positions. The alignment results were then used as input for Cufflinks (Version 2.1.1, RRID:SCR_014597) [47] to assemble the transcripts into the gene models. In addition, RNA-seq data were assembled using Trinity (Version 2.1.1, RRID:SCR_013048) [48], creating several pseudo-ESTs (short for expressed sequence tags). These pseudo-ESTs were also mapped to the assembled genome using BLAT [49] (RRID:SCR_011919), and gene models were predicted using PASA [50] (RRID:SCR_014656). A weighted and non-redundant gene set was generated using EVidenceModeler (EVM, Version 1.1.1, RRID:SCR_014659) [51], which merged all the gene models predicted by the above three approaches. Finally, PASA was used to adjust the gene models generated by EVM. As a result, 39,733 protein-coding genes were annotated in our final set. To better understand gene functions, we used all our 39,733 protein-coding genes as queries against public protein databases, including NCBI non-redundant protein sequences, Swiss-Prot, Protein family, Kyoto Encyclopedia of Genes and Genomes (KEGG), InterPro, and Gene Ontology (GO). In total, 39,646 genes (99.8%) were annotated using these databases, and 24,402 (61.41%) were supported by RNA-seq data. Furthermore, the number of genes, the gene length distribution, and the exon length distribution were all comparable to those of other maize inbred lines and common crop species (Table 3).

Transfer RNA (tRNA) genes were predicted using tRNAscan-SE (Version 1.4, RRID:SCR_010835) [52] with the default parameters. Ribosomal RNAs (rRNAs) were annotated based on their homology levels with the rRNAs of several species of higher plants using BLASTN with an *E*-value of $1 \times 10^{-5}$. The microRNA (miRNA) and small nuclear RNA (snRNA) fragments were identified by searching the Rfam database (Version 11.0, RRID:SCR_007891) using Infernal (Version 1.1, RRID:SCR_011809) [53, 54]. Finally, 4547 miRNAs, 5963 tRNAs, 63,564 rRNAs, and 1422 snRNAs were identified, with average lengths of 126.79, 75.25, 309.47, and 132.10 bp, respectively (Table 4).

## Comparative genomic analysis between Dan340 and other maize lines

We applied the OrthoMCL pipeline [55] to identify orthologous gene families among the four maize inbred lines, including Dan340, B73, Mo17, and SK. The longest protein from each gene was selected, and the proteins with a length of less than 30 amino acids were removed. Subsequently, pairwise sequence similarities between all input protein sequences were

**Table 4.** Annotation statistics of the non-coding RNAs in the Dan340 genome using different databases.

|  |  | Type | Copy (w*) | Average length (bp) | Total length (bp) |
|---|---|---|---|---|---|
| rRNA | miRNA | 4547 | 126.79 | 576,516 | 0.024546 |
|  | tRNA | 5963 | 75.25 | 448,705 | 0.019104 |
|  | rRNA | 63,564 | 309.47 | 19,671,118 | 0.84 |
|  | 18S | 6607 | 1727.38 | 11,412,778 | 0.49 |
|  | 28S | 25,188 | 143.61 | 3,617,315 | 0.15 |
|  | 5.8S | 25,181 | 153.48 | 3,864,710 | 0.16 |
|  | 5S | 6588 | 117.84 | 776,315 | 0.033053 |
| snRNA | snRNA | 1,422 | 132.1 | 187,845 | 0.007998 |
|  | CD-box | 647 | 103.2 | 66,768 | 0.002843 |
|  | HACA-box | 123 | 126.27 | 15,531 | 0.000661 |
|  | splicing | 651 | 161.72 | 105,278 | 0.004482 |

calculated using BLASTP [56] (RRID:SCR_001010) with an $E$ value cut-off of $1 \times 10^{-5}$. Markov clustering (MCL) of the resulting similarity matrix was used to define the ortholog cluster structure of the proteins, using an inflation value (-$I$) of 1.5 (default setting of OrthoMCL).

Next, comparative analyses were performed among Dan340, B73, Mo17, and SK (Figure 7A). The genes from the Dan340 genome and those from B73, Mo17 and SK were clustered into 27,654 gene families. Of these, 15,690 families were shared among the four maize inbred lines, representing a core set of genes across these maize genomes. We found 1806 genes from 359 gene families that were specific to Dan340, of which many had functional GO annotations related to "protein phosphorylation", "single-organism catabolic process", and "pheromone binding" (Figure 7B). Using the KEGG functional enrichment, the most enriched pathways of the Dan340-specific genes were "antifolate resistance", "epithelial cell signaling in *Helicobacter pylori* infection", and "pentose and glucuronate interconversions" (Figure 7C). In addition, OrthoMCL was used to identify the core and dispensable gene sets based on gene families. The gene families that were shared among the four inbred lines were defined as core gene families. Furthermore, gene families shared among three inbred lines, between two inbred lines, and those only present in one inbred line (private gene families) are also displayed in Figure 7D.

## Genetic variation analysis

To investigate the genetic and structural variations between Dan340 and other maize inbred lines, we first aligned the other three genomes to the Dan340 reference genome based on MUMmer (V 4.0.0 beta2) [57] (RRID:SCR_018171) with parameter "—mum -g 1000 -c 90 -l 40". Then the alignment files were filtered to generate 1-to-1 mapping by delta-filter with parameters:"-m -i 90 -l 100". The genomes of B73 and Mo17 were downloaded from MaizeGDB [58], and the genome of SK was obtained from the National Genomics Data Center [59]. Next, the output of Nucmer was analyzed using SyRI [60] with default parameters to identify variation. On the basis of the above pipeline, we obtained structural variation sets and generated into the vcf file. We also used PBSV (Version 2.2.2) [61] to investigate the genetic and structural variations (details and outputs available via the GigaDB entry [62]).

The high-quality Dan340 reference genome allowed us to identify large SVs in different maize inbred lines. By aligning the genome of B73 to the Dan340 genome, we identified 36,363 structural variations (longer than 500 bp) between the two representative maize genomes, including 15,923 insertions, 16,173 deletions, 141 inversions, and 4,126

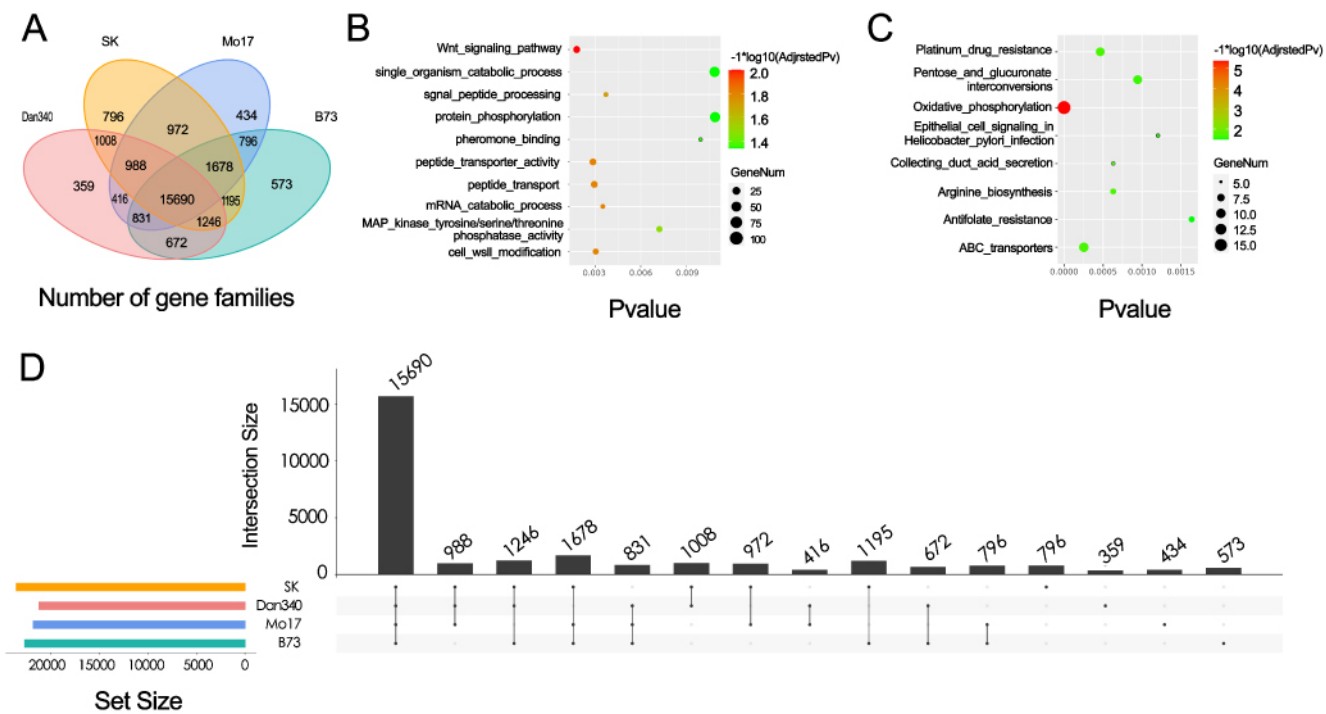

**Figure 7.** Gene family analyses and core- and pan-genomes of maize. (A) Comparisons of gene families in Dan340, B73, Mo17, and SK. The Venn diagram illustrates the shared and unique gene families among the four maize inbred lines. (B) GO enrichment analysis of Dan340-specific genes. (C) KEGG analysis of Dan340-specific genes. (D) Core- and pan-genomes of maize. The histograms show the core-gene clusters (shared by all four genomes), dispensable gene clusters (present in three or two genomes), and specific gene clusters (present only in one genome).

**Table 5.** Structural variations between Dan340 and B73.

| Chr. Number | Insertion | | Deletion | | Inversion | | Duplication | |
|---|---|---|---|---|---|---|---|---|
| | Number | Length (bp) | Number | Length (bp) | Number | Length (bp) | Number | Length (bp) |
| 1 | 2,196 | 17,315,745 | 2,352 | 18,827,437 | 20 | 1,157,776 | 456 | 1,574,417 |
| 2 | 1,969 | 21,308,887 | 2,037 | 24,356,338 | 27 | 44,078,181 | 885 | 6,684,632 |
| 3 | 1,908 | 16,163,199 | 1,951 | 16,640,435 | 14 | 13,164,411 | 424 | 1,376,864 |
| 4 | 1,764 | 14,082,107 | 1,803 | 15,310,877 | 17 | 1,036,816 | 338 | 1,118,242 |
| 5 | 1,395 | 11,820,766 | 1,400 | 11,943,768 | 12 | 7,848,883 | 331 | 1,422,310 |
| 6 | 1,237 | 12,859,269 | 1,179 | 12,589,548 | 11 | 12,425,596 | 368 | 1,291,481 |
| 7 | 1,513 | 15,084,605 | 1,463 | 16,126,941 | 7 | 19,727,655 | 534 | 6,342,568 |
| 8 | 1,442 | 12,042,545 | 1,420 | 11,464,697 | 12 | 531,876 | 301 | 1,034,272 |
| 9 | 1,397 | 13,366,945 | 1,421 | 13,770,696 | 9 | 3,773,691 | 286 | 992,215 |
| 10 | 1,102 | 8,984,895 | 1,147 | 9,437,955 | 12 | 362,594 | 203 | 692,374 |

duplications (Table 5). Furthermore, the structural variations presented in Mo17 and SK were also detected in this study (Tables 6 and 7). The dataset generated by PBSV is available in GigaDB [62]. These datasets provide abundant variation resources for future molecular improvements and breeding in maize.

## CONCLUSIONS

We assembled the chromosome-level genome of the maize elite inbred line Dan340 using long CCS reads from the third-generation PacBio Sequel II sequencing platform, with scaffolding informed by Hi-C. The final assembly of the Dan340 genome was 2348.72 Mb,

**Table 6.** Structural variations between Dan340 and Mo17.

| Chr. Number | Insertion | | Deletion | | Inversion | | Duplication | |
|---|---|---|---|---|---|---|---|---|
| | Number | Length (bp) | Number | Length (bp) | Number | Length (bp) | Number | Length (bp) |
| 1 | 2,112 | 16,595,761 | 2,268 | 18,826,385 | 24 | 1,892,611 | 364 | 1,360,243 |
| 2 | 1,775 | 17,433,432 | 1,822 | 19,928,356 | 27 | 44,197,495 | 671 | 3,764,856 |
| 3 | 1,810 | 15,987,579 | 1,850 | 16,785,758 | 11 | 12,931,342 | 364 | 1,093,740 |
| 4 | 1,305 | 10,393,408 | 1,360 | 12,151,836 | 12 | 8,399,292 | 270 | 932,811 |
| 5 | 1,401 | 11,584,434 | 1,481 | 12,571,981 | 13 | 6,688,118 | 292 | 961,542 |
| 6 | 1,163 | 11,784,318 | 1,182 | 12,742,685 | 18 | 13,228,176 | 382 | 1,182,254 |
| 7 | 1,466 | 15,144,929 | 1,515 | 18,539,394 | 6 | 19,813,930 | 520 | 4,613,073 |
| 8 | 1,266 | 9,520,525 | 1,386 | 11,021,444 | 15 | 1,387,813 | 290 | 1,134,832 |
| 9 | 1,335 | 12,303,841 | 1,388 | 13,598,480 | 13 | 4,032,985 | 282 | 1,024,261 |
| 10 | 1,083 | 8,801,874 | 1,138 | 9,738,720 | 17 | 1,044,530 | 247 | 812,820 |

**Table 7.** Structural variations between Dan340 and SK.

| Chr. Number | Insertion | | Deletion | | Inversion | | Duplication | |
|---|---|---|---|---|---|---|---|---|
| | Number | Length (bp) | Number | Length (bp) | Number | Length (bp) | Number | Length (bp) |
| 1 | 2,242 | 16,991,522 | 2,435 | 19,097,136 | 33 | 2,305,401 | 557 | 1,959,780 |
| 2 | 1,956 | 21,684,186 | 2,045 | 24,111,211 | 23 | 50,179,678 | 923 | 5,280,910 |
| 3 | 1,909 | 15,947,617 | 1,943 | 15,909,065 | 20 | 13,717,305 | 450 | 1,485,303 |
| 4 | 2,026 | 16,759,290 | 2,101 | 17,565,665 | 28 | 3,102,314 | 390 | 1,238,308 |
| 5 | 1,665 | 13,409,385 | 1,727 | 14,676,710 | 28 | 8,960,125 | 381 | 1,429,527 |
| 6 | 1,201 | 12,554,766 | 1,251 | 12,023,404 | 12 | 15,291,801 | 434 | 1,552,484 |
| 7 | 1,502 | 15,584,044 | 1,554 | 16,932,500 | 17 | 19,884,384 | 609 | 2,554,084 |
| 8 | 1,229 | 9,421,014 | 1,311 | 10,010,279 | 8 | 398,434 | 324 | 1,110,257 |
| 9 | 1,787 | 18,944,180 | 1,866 | 17,898,278 | 13 | 3,443,024 | 362 | 1,285,568 |
| 10 | 1,087 | 8,296,044 | 1,189 | 9,755,942 | 15 | 966,661 | 260 | 870,594 |

including 2738 contigs and 2315 scaffolds with N50 of 41.49 Mb and 215.35 Mb, respectively. Comparisons of the Dan340 genome with the reference genomes of three other common maize inbred lines identified 1806 genes from 359 gene families that were specific to Dan340. In addition, we also obtained large numbers of structural variants between Dan340 and other maize inbred lines, and these may be underlying the mechanisms responsible for the phenotypic discrepancies between Dan340 and other maize varieties. Therefore, the assembly and annotation of this genome improves our understanding of the intraspecific genomic diversity in maize and provides novel resources for maize breeding improvements.

## DATA AVAILABILITY

The raw sequence data have been deposited in NCBI under project accession No. PRJNA795201. Data is also available in the *GigaScience* GigaDB repository [62].

## ABBREVIATIONS

BUSCO: Benchmarking Universal Single-Copy Orthologs; Hi-C: chromosomal conformation capture; CCS: circular consensus sequencing; EVM: EVidenceModeler; HiFi: long high-fidelity; LTR: long-terminal repeat; LAI: long-terminal repeat assembly index; TEs: transposable elements; EVM: EVidenceModeler; KEGG: Kyoto Encyclopedia of Genes and Genomes; GO: Gene Ontology; MCL: Markov clustering; NCBI: National Center for Biotechnology Information; nucleotides (N); PacBio: Pacific Biosciences; SMRT: single-molecule real-time; SV: structural variations; tRNA: Transfer RNA; rRNAs: Ribosomal RNAs; miRNA: microRNA; snRNA: small nuclear RNA.

## COMPETING INTERESTS

The authors declare that they have no competing interests.

## AUTHORS' CONTRIBUTIONS

FW, JZ and HZ conceived the project; Y-KZ, DM and YW wrote and modified the manuscript; LX, GF and LW performed the data curation; YH, LZ, Y-LZ and ZL analyzed the data. All authors read and approved the final manuscript.

## FUNDING

This research was supported by grants from the special project for the construction of scientific and technological innovation capacity of Beijing Academy of Agriculture and Forestry Sciences (NO. KJCX20200305).

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
