## [Reviewer Report]

Comments on revised manuscriptThe authors have done due diligence in addressing the reviewers comments. Manuscript is acceptable.

---

## [Reviewer Report]

Reviewer name and names of any other individual's who aided in reviewer Kapeel ChouguleDo you understand and agree to our policy of having open and named reviews, and having your review included with the published papers. (If no, please inform the editor that you cannot review this manuscript.)YesIs the language of sufficient quality?YesPlease add additional comments on language quality to clarify if needed
Are all data available and do they match the descriptions in the paper? NoAdditional CommentsI typed Bioproject ID provided under supporting data and materials: PRJNA795201 but could not see any information or data.

Either the authors have put it on hold or data have not be submitted. Please request the authors to release data.Are the data and metadata consistent with relevant minimum information or reporting standards? See GigaDB checklists for examples <a href="http://gigadb.org/site/guide" target="_blank">http://gigadb.org/site/guide</a>YesAdditional CommentsIs the data acquisition clear, complete and methodologically sound?YesAdditional CommentsIs there sufficient detail in the methods and data-processing steps to allow reproduction?YesAdditional CommentsIs there sufficient data validation and statistical analyses of data quality? YesAdditional CommentsIs the validation suitable for this type of data?YesAdditional CommentsIs there sufficient information for others to reuse this dataset or integrate it with other data?YesAdditional CommentsAny Additional Overall Comments to the AuthorThe manuscript titled "A chromosome-level genome assembly and annotation of a maize elitebreedinglineDan340" provides a good overview of genome assembly construction and structural annotation of maize elite breeding line Dan340. The authors have presented correct methods in construction of genome assembly and annotations. Although the paper provides elaborate methods for genome construction the manuscript lacking to demonstrate the value of the genome as a resource. More specifically the authors describe this line as elite with having desirable characters such as disease resistance; lodging resistance and so on. The focus of the manuscript mostly on methods without significant examples to demonstrate the value of the resource. The authors could characterize some disease resistance genes or genes affected by structural variations in the Dan340 line and compare it to other maize lines.

Major Comments: 
1) I typed Bioproject ID provided under supporting data and materials: PRJNA795201 but could not see any information or data. Either the authors have put it on hold or data have not be submitted. Please request the authors to release data.
2) The authors build repeat lib using ab inito and homology based methods and masked 66.09% of the Dan340 genome; this when compared to other reference lines esp B73 from Hufford et al (https://www.science.org/action/downloadSupplement?doi=10.1126%2Fscience.abg5289&file=science.abg5289_hufford_sm.pdf) Table S5 is significatly lower i.e B73 genome is 85% masked. 
3) To assess the quality of genome assembly the authors use LAI index. The reported LAI for B73 in Hufford et al ( same as above Table S2) is 27.84 where as the authors report B73 LAI 16.79 which is incorrect. Is the B73 version used for comparison v4 or v5??. Can the authors provide a pairwise comparison of genome sequence using dot plot betwee Dan340 line and other maize lines to visualize assembly artifacts like inversions deletions or gaps in the assembly.
4) The authors use transcription data from six tissues ( stem,endosperm,embryo,bract,silk,ear and tip) for alignment. There is no mention in the manuscript how these were generated. Are they also submitted to NCBI?

Minor comments: 
1) Line 6; rephrase sentence what the authors meant: There are more than 50 Maize hybrid breeds derived from Dan340 since 2000.
2) Table 2 : use full genus specie names in column 1
RecommendationMajor Revision

---

## [Reviewer Report]

Reviewer name and names of any other individual's who aided in reviewer Georg HabererDo you understand and agree to our policy of having open and named reviews, and having your review included with the published papers. (If no, please inform the editor that you cannot review this manuscript.)YesIs the language of sufficient quality?YesPlease add additional comments on language quality to clarify if needed
Are all data available and do they match the descriptions in the paper? NoAdditional CommentsI could not check the data availability, the provided project number was not at the NCBI sequence archive. The authors should ensure that both raw genomic and RNAseq reads are uploaded there. Also, the final genome sequence and gene annotations should be available to the community.Are the data and metadata consistent with relevant minimum information or reporting standards? See GigaDB checklists for examples <a href="http://gigadb.org/site/guide" target="_blank">http://gigadb.org/site/guide</a>YesAdditional CommentsIs the data acquisition clear, complete and methodologically sound?YesAdditional CommentsIs there sufficient detail in the methods and data-processing steps to allow reproduction?NoAdditional CommentsThe methods how structural variants were detected is very fuzzy. For example, L283: “Then, SAMtools v0.1.1 (Li et al., 2009) was used to assign the structural variations from the bam file to each chromosome or scaffold.” Samtools per se does not call any SVs, it is unclear how this assignment was actually performed: did they filter SV here? Did they use the full samtools+mpileup+vcftools pipeline? Also, did they really apply v0.1.1, this is an extremely outdated version (current version 1.15, probably dozens of updates)Is there sufficient data validation and statistical analyses of data quality? YesAdditional CommentsIs the validation suitable for this type of data?YesAdditional CommentsIs there sufficient information for others to reuse this dataset or integrate it with other data?YesAdditional CommentsAny Additional Overall Comments to the AuthorThe authors report sequencing and assembly of Dan340, a highly important founder line for maize breeding in China. They complement the genome sequence by gene and repeat annotations and a preliminary study of structural variants between their and three other lines. The obtained genome sequence is of excellent quality and evaluated by several independent statistics (BUSCO, CEGMA, LAI). Repeat and gene predictions seem to be done by state-of-the-art methods, and the reported numbers and proportions are similar to previous reports and comparative analysis in maize. In summary, the manuscript provides a highly valuable genomic resource for maize biologists and breeders and complements the increasing number of maize pan-genomes by a major Chinese germplasm. 
I have only a few comments for the authors:
see my points above about data availability and methods to call SVs, in addition:
- Table 2: they provide here only abbreviations of species, they have to spell out these abbreviations in the table legend, for exampe: Hvu: Hordeum vulgare. Also it may be difficult for readers to understand to what species/lines ‘ZmaL’ and ‘Zma’ point.

- L31/L67: “… and so on.” not an appropriate closure of a sentence in scientific texts.
- L95-103: this part can be left out, the authors do not have to, and should not describe or even justify here the CCS technology. Just mention what has been done.

RecommendationMinor Revision

---

## [Reviewer Report]

Reviewer name and names of any other individual's who aided in reviewer Xupo DingDo you understand and agree to our policy of having open and named reviews, and having your review included with the published papers. (If no, please inform the editor that you cannot review this manuscript.)YesIs the language of sufficient quality?YesPlease add additional comments on language quality to clarify if needed
Are all data available and do they match the descriptions in the paper? YesAdditional CommentsAre the data and metadata consistent with relevant minimum information or reporting standards? See GigaDB checklists for examples <a href="http://gigadb.org/site/guide" target="_blank">http://gigadb.org/site/guide</a>YesAdditional CommentsIs the data acquisition clear, complete and methodologically sound?YesAdditional CommentsIs there sufficient detail in the methods and data-processing steps to allow reproduction?YesAdditional CommentsIs there sufficient data validation and statistical analyses of data quality? YesAdditional CommentsIs the validation suitable for this type of data?YesAdditional CommentsIs there sufficient information for others to reuse this dataset or integrate it with other data?YesAdditional CommentsAny Additional Overall Comments to the Author1. The gene numbers and repeat percentage should be presented in the abstract.
2. The potential functions of three secondary metabolite processes in the abstract might be inferred. It will be showed specifics of Dan340, such as related to disease resistance or others?
3. Line 79-82, this sentence is same with the conclusion in the abstract, please extending.
4. The depth or data size of CCS and Hi-C should be added to corresponding the Illumina data description. 
5. Line 128-131, before assembly assessment, the quality might not be steerable. Please consulting: The assembly was performed in a stepwise fashion with PacBio HiFi reads and Illumina short reads and Hi-C technology.
6. Line 211-213, insert the description about comparative data of LTR in Dan340, B73, Mo17, and SK.
7. Line 241, describe how many genes or percentage of protein coding genes were supported by RNA-seq.
8. Fig.4A, the line name of four maize might be out of Veen.
9. Fig.4B and Fig.4C were not cited in the data description.
10. Line 264-270, insert the pathways description fronting five or ten in the GO list and infer the function of special three secondary metabolite pathway in Dan340.
11. In conclusion, the contributions should be deepen discussion，at least not exactly same with details in abstract and Line 79-82.RecommendationMinor Revision